# Vitamin D Receptor Gene Expression in Adipose Tissue of Obese Individuals is Regulated by miRNA and Correlates with the Pro-Inflammatory Cytokine Level

**DOI:** 10.3390/ijms20215272

**Published:** 2019-10-24

**Authors:** Marta Izabela Jonas, Alina Kuryłowicz, Zbigniew Bartoszewicz, Wojciech Lisik, Maurycy Jonas, Krzysztof Kozniewski, Monika Puzianowska-Kuznicka

**Affiliations:** 1Department of Human Epigenetics, Mossakowski Medical Research Centre, Polish Academy of Sciences, 5 Pawinskiego Street, 02-106 Warsaw, Poland; martajonas@poczta.onet.pl (M.I.J.); zbigniew.bartoszewicz@wum.edu.pl (Z.B.); krzychukoz@gmail.com (K.K.); mpuzianowska@imdik.pan.pl (M.P.-K.); 2Department of Internal Diseases and Endocrinology, Medical University of Warsaw, 1a Banacha St, 02-097 Warsaw, Poland; 3Department of General and Transplantation Surgery, Medical University of Warsaw, 00-001 Warsaw, Poland; wojciech.lisik@wum.edu.pl (W.L.); morjon@poczta.onet.pl (M.J.); 4Department of Geriatrics and Gerontology, Medical Centre of Postgraduate Education, 01-826 Warsaw, Poland

**Keywords:** adipose tissue, obesity, vitamin D receptor (VDR), micro RNA (miRNA), pro-inflammatory cytokines

## Abstract

*Background*: Given the role that vitamin D (VD) plays in the regulation of the inflammatory activity of adipocytes, we aimed to assess whether obesity changes the expression of VD-related genes in adipose tissue and, if so, to investigate whether this phenomenon depends on microRNA interference and how it may influence the local inflammatory milieu. *Methods*: The expression of genes encoding VD 1α-hydroxylase (*CYP27B1*), 24-hydroxylase (*CYP24A1*) and receptor (*VDR*), selected interleukins and microRNAs was evaluated by real-time PCR in visceral (VAT) and in subcutaneous (SAT) adipose tissues of 55 obese (BMI > 40 kg/m^2^) and 31 normal-weight (BMI 20–24.9 kg/m^2^) individuals. *Results*: *VDR* mRNA levels were higher, while *CYP27B1* levels were lower in adipose tissues of obese patients than in those of normal-weight controls (VAT: *P* = 0.04, SAT: *P* < 0.0001 and VAT: *P* = 0.004, SAT: *P* = 0.016, respectively). The expression of *VDR* in VAT of obese subjects correlated negatively with levels of miR-125a-5p (*P* = 0.0006, r_s_ = −0.525), miR-125b-5p (*P* = 0.001, r_s_ = −0.495), and miR-214-3p (*P* = 0.009, r_s_ = −0.379). Additionally, *VDR* mRNA concentrations in visceral adipose tissues of obese subjects correlated positively with mRNA levels of interleukins: 1β, 6 and 8. *Conclusions*: We observed obesity-associated up-regulation of *VDR* and down-regulation of *CYP27B* mRNA levels in adipose tissue. *VDR* expression correlates with the expression of pro-inflammatory cytokines and may be regulated by miRNAs.

## 1. Introduction

The discovery of vitamin D receptor (VDR) and its identification in a wide panel of tissues broadened our understanding about the role of vitamin D_3_ (cholecalciferol, hereinafter referred to as vitamin D, VD) in the maintenance of body homeostasis. In the context of adipose tissue, active forms of VD were found to participate in the regulation of lipogenesis and adipogenesis, as well as adipokines’ production and secretion; however, these effects are species-specific [1,2,3]. 

In human mesenchymal progenitor cells and preadipocytes isolated from the subcutaneous adipose tissue (SAT) active form of VD, calcitriol (1,25(OH)_2_D_3_) was found to promote differentiation toward mature adipocytes, enhance lipid accumulation and inhibit adiponectin secretion [4,5,6]. VD metabolites also play a crucial role in the inflammatory activity of human adipocytes; however, results of the available studies are not consistent. In a micro-array study, calcitriol, via regulation of almost 140 genes, favoured inflammation and oxidative stress in mature human subcutaneous adipocytes [7]. In turn, in cultures of pre-adipocytes, the addition of 1,25(OH)_2_D_3_ down-regulated the synthesis of monocyte chemoattractant protein-1 (MCP1), interleukins (IL) 1β, 6 and 8 directly (via interaction of VDR with VD-responsive elements (VDRE) located in the regulatory regions of their genes), as well as by inhibition of the pro-inflammatory nuclear factor κB (NF-κB) pathway [5,8,9,10].

In the course of obesity, the accumulation of lipids leads to the increased expression of genes encoding cytokines, chemokines and adhesion molecules in adipocytes, leading to the infiltration of immune cells that produce further pro-inflammatory mediators [11]. This shift in the secretory profile of adipose tissue contributes to the development of obesity-related metabolic and cardiovascular complications [12]. Increased levels of pro-inflammatory adipokines circulating in the blood impair endothelial function and lead to the thickening of the vascular smooth muscle layer. These subsequently result in increased oxidative stress, stiffening of the vascular wall and impaired vessel reactivity [13]. The unfavourable effect of adipokines on the cardiovascular system can also be exerted indirectly, via other organs and systems, e.g., obesity-associated high leptin serum levels increase blood pressure by the stimulation of certain pathways in the central nervous system and by enhancing the sympathetic nervous system output [12].

Given the role that VD plays in the regulation of the inflammatory activity of adipocytes, it was plausible that low-VD status may be one of the triggers of the inflammatory state in adipose tissue. However, while several authors confirmed a link between obesity and chronic inflammation, studies regarding the association of vitamin D status with the body mass index (BMI) and body composition are equivocal. In some populations, 25(OH)D_3_ levels had been inversely correlated with BMI, fat mass and adipose tissue distribution, while in other populations, no evident correlations between those parameters were found (reviewed in [14]). It was also suggested that not VD status itself, but local activity of vitamin D hydroxylases: 1α-hydroxylase (its principal form CYP27B1), responsible for VD activation, and 24-hydroxylase (CYP24A1), responsible for VD degradation (Figure 1) and availability of VDR in adipose tissue, may be crucial for the modulation of the inflammatory activity of adipocytes by calcitriol [2]. 

The expressional activity of *CYP27B1*, *CYP24A1* and *VDR* genes, all possessing VDRE in their regulatory regions, can be modulated by VD itself, but also by other factors which play important roles in epigenetic modifications, such as microRNA (miRNA) interference [15,16,17]. In this study, we aimed to evaluate if obesity influences the expression of genes crucial for VD activation (*CYP27B1*), inactivation (*CYP24A1*) and action (*VDR*) in different adipose tissues depots. Next, we investigated if VD status and miRNA interference could play a role in this phenomenon. Finally, by correlating the mRNA levels of these genes with mRNA levels of the pro-inflammatory cytokines, we aimed to assess how obesity-related changes in VD-related genes expression may impact the local inflammatory milieu in adipose tissue. 

## 2. Results

### 2.1. VDR and CYP27B1 Have Opposite Expression Profiles in Adipose Tissues 

*VDR* mRNA levels (Figure 2a) were significantly higher in obese subjects, compared to normal-weight controls, both in visceral (*P* = 0.04) and in subcutaneous (*P* < 0.0001) adipose tissues. On the contrary, mean mRNA levels for *CYP27B1* (Figure 2b) were lower in obese individuals (*P* = 0.004 for VAT and *P* = 0.016 for SAT). In both the case of *VDR* and of *CYP27B1*, we observed no differences in their mRNA levels between different adipose tissue depots in obese and normal-weight subjects. In contrast, the mean mRNA levels of *CYP24A1* did not differ significantly between the investigated tissues (Figure 2c)

### 2.2. VDR and CYP27B1 Expression Profiles Do Not Depend on Vitamin D Metabolites’ Serum Levels

According to the vitamin D supplementation guidelines for the polish population, 87.9% (51 out of 55) of the obese individuals and 64.52% (20 out of 31) of the normal-weight subjects were VD-deficient (25(OH)D_3_ level < 20 ng/mL) [18].

Mean 25(OH)D_3_ levels in the obese study participants did not differ significantly from those measured in the normal-weight controls (15.86 ng/dL vs. 16.59 ng/dL, Table 1). Similarly, we found no significant differences in the mean 1,25(OH)_2_D_3_ levels between the studied groups (130.41 pmol/L in obese subjects vs. 133.99 pmol/L in the normal-weight individuals). 

To determine if VD status might play a role in the regulation of *VDR* and *CYP27B1* expression in adipose tissue, we correlated 25OHD_3_ and 1,25(OH)_2_D_3_ serum concentrations with *VDR* and *CYP27B1* mRNA levels in the investigated tissues. However, we did not observe any significant correlations either in the obese individuals or in the control group (data not shown).

### 2.3. Expression of the Selected miRNAs Correlates Negatively with VDR mRNA Levels in Adipose Tissues

Finding no correlations between the VD metabolites serum concentrations with *VDR* and *CYP27B1* mRNA levels in the investigated tissues, we decided to investigate whether miRNAs play a role in the regulation of these two genes’ expression. Using bioinformatics tools (TargetScanHuman available at http://www.targetscan.org, miRanda-mirSVR available at http://www.microrna.org/microrna/home.do and the Pictar available at http://pictar.mdc-berlin.de), results of the next-generation-sequencing screening [19] and based on the available literature covering in vitro studies, we selected miRNA that potentially interfere with *VDR* and *CYP27B1* 3′UTR sequences (Appendix A). Next, we measured their concentrations in the investigated tissues in order to correlate them with *VDR* and *CYP27B1* mRNA levels.

Among several possible microRNAs that potentially interact with the *VDR* 3′UTR sequence, we selected hsa-miR-125a-5p, hsa-miR-125b-5p and hsa-miR-214-3p for analysis. In the case of all these miRNAs, we found significant differences in their expression between the investigated tissues (Appendix A and Kurylowicz et al., 2016 and 2017 [19,20]) and their role in the regulation of *VDR* expression was confirmed by in vitro studies [15,21,22]. 

In obese individuals, we observed significant negative correlations between *VDR* mRNA level and hsa-miR-125a-5p (*P* = 0.0006, r_s_ = −0.525, Figure 3a), hsa-miR-125b-5p (*P* = 0.001, r_s_ = −0.495, Figure 3b) and hsa-miR-214-3p (*P* = 0.009, r_s_ = −0.379, Figure 3c) in VAT. However, we did not observe similar correlations in SAT of obese individuals (Figure 3d–f) or in tissues obtained from the normal-weight subjects (Appendix A).

Among miRNAs targeting *CYP27B1*, expression of hsa-miR-22-3p and hsa-miR-450b-5p differed significantly between the investigated tissues (Appendix A, Kurylowicz A, 2016 and 2017 [19,20]. However, we observed no significant correlations between the expression of these miRNA and *CYP27B1* mRNA levels (Appendix A).

### 2.4. VDR mRNA Levels Correlate Positively with the Expression of Genes Encoding Pro-Inflammatory Cytokines in Adipose Tissues of Obese Individuals

Next, we investigated the relationship between the *VDR* mRNA level and expression of genes encoding pro-inflammatory cytokines for which expression is, via VDR, regulated by 1,25(OH)_2_D_3_. In obese patients, we found positive correlations between *VDR* mRNA levels and levels of mRNA for genes encoding *IL-6* (in VAT-O: *P* = 0.009, r_s_ = 0.471, Figure 4a, in SAT-O: *P* = 0.01, r_s_ = 0.512, Figure 4b); *IL-8* (in VAT-O: *P* = 0.049, r_s_ = 0.439, Figure 4c, in SAT-O: *P* = 0.021, r_s_ = 0.449, Figure 4d) and *IL-1β* (in VAT-O: *P* < 0.0001, r_s_ = 0.714, Figure 4e). 

## 3. Discussion

In this work, we investigated whether excessive adiposity in humans is associated with changes in the expression of VD-related genes in adipose tissue and looked for the possible mechanisms responsible for this phenomenon. Next, by correlating the expression of the investigated genes with mRNA levels of VD-dependent pro-inflammatory cytokines, we assessed how the altered expression of key genes involved in VD metabolism and action might influence the inflammatory activity of adipose tissue in obese individuals.

Epidemiological studies suggest that obese individuals are more vulnerable to VD deficiency than those of normal weight. An unhealthy diet, low in valuable nutrients, leading to relative malnutrition despite a high body mass, may explain this phenomenon [14]. Moreover, it has been proposed that a low-VD status in obese subjects results from the sequestration of VD derivates in adipose tissue [23]. Despite the general trend, in our group, we observed no significant differences in VD metabolites levels between the obese and normal-weight individuals. There can be several reasons for this finding. Firstly, VD deficiency (defined as 25(OH)D_3_ level below 20 ng/mL) is relatively common and affects approximately 90% of the entire Polish population [18]. Secondly, some technical issues could exist regarding the processing of the sera that might have influenced the results, e.g., exposition to sunlight or high temperatures. However, the sera were collected with the intention of measuring the concentration of VD metabolites and stored in the dark, in non-permeable tubes. 

Since VD acts as an important regulator of the inflammatory activity of adipocytes, a theory that low-VD status may contribute to the development of chronic inflammation in obesity was plausible. Finding that the obese study participants do not differ significantly from normal-weight individuals in terms of 25(OH)D_3_ and 1,25(OH)_2_D_3_ serum levels led us to the idea, that not VD status itself, but local disturbances in VD metabolism and action may be responsible for the increased inflammatory activity of adipose tissue in obesity.

To verify this hypothesis, we compared the expression of VD-related genes in adipose tissues of obese and normal-weight subjects. We found that obesity was associated with increased *VDR* mRNA levels in adipose tissues, while in the case of *CYP27B1*, an opposite trend was observed. Our results regarding the obesity-related increased expression of *VDR* in adipose tissues are coherent with data from previous studies in morbidly obese individuals [24,25]. Moreover, Wamberg et al., in a study performed in 20 obese and 20 normal-weight subjects, showed the same pattern of *CYP27B1* and *CYP24* expression as that observed in our cohort [8]. Lower expression of *CYP27B1* in adipose tissues of obese subjects might translate to lower enzyme activity and subsequently, to the lower local concentrations of the active VD metabolite. Therefore, higher *VDR* expression may represent a compensation mechanism in response to the synthesis of low amounts of 1,25(OH)_2_D_3_. Interestingly, in obese individuals, we did not observe significant differences in *VDR* and *CYP27B1* expression between the different adipose tissue depots. This finding may reflect the true phenomenon or result from the fact that our measurements were performed on the mRNA level and might not translate to the protein level.

Both 25(OH)D_3_ and 1,25(OH)_2_D_3_ induced expression of VD-related genes in cultures of primary human adipocytes [4]. There are also studies showing the influence of serum 25(OH)D_3_ and 1,25(OH)_2_D_3_ concentrations on *VDR* and *CYP27B1* mRNA levels in adipose tissue [8,24]. In the studied population, neither *VDR* nor *CYP27B1* mRNA levels in adipose tissues correlated with VD metabolites’ concentration in serum. Measurement of VD metabolites directly in adipose tissues extracts would have been more informative in this aspect; however, the methodology is highly complicated, and despite numerous efforts, we failed to perform these measurements (data not shown). 

Upon finding that *VDR* and *CYP7B1* mRNA levels in adipose tissue do not correspond with the VD status, we looked for other factors that could possibly influence the expression of these two genes. Previous studies have shown that epigenetic modifications, e.g., the methylation of regulatory regions, may influence the expression of VD-related genes in adipose tissues [26]. Given the proven role of miRNA in adipose tissue metabolism [27], as well as in the regulation of *VDR* and *CYP27B1* expression in other tissues [15,16,28], we investigated whether miRNA interference may be involved in this case. We found significant, negative correlations between *VDR* mRNA level and levels of hsa-miR-125a-5p, hsa-miR-125b-5p, and hsa-miR-214-3p in VAT of obese subjects. All these miRNAs were found to participate in the regulation of *VDR* expression in vitro [15,21,22], and their expression is decreased in visceral adipose tissue in obesity [19,20]. We do not observe similar correlations in SAT of obese individuals. Given the numerous differences in metabolism and secretional activity between VAT and SAT, the mechanisms controlling genes expression in these two adipose tissue depots might also be distinct. In the case of *VDR*, its tissue-specific expression may result, e.g., from the regulation by different enhancers present upstream of the promoter [29] or different methylation levels in the regulatory regions [26]. The finding that none of the investigated miRNA correlated with *CYP27B1* mRNA levels suggests that other mechanisms (e.g., those related to insulin resistance) must be responsible for the decreased *CYP27B1* expression in the adipose tissue in obese subjects [30].

Finally, we decided to investigate how increased *VDR* expression in the adipose tissue of obese individuals may influence its inflammatory activity. For this purpose, we correlated *VDR* mRNA levels with the mRNA levels for genes encoding IL-1β, IL-6 and IL-8—all possessing VDRE in their regulatory regions. Expression of the three investigated interleukins correlated positively with the *VDR* mRNA levels in VAT of obese individuals, while *IL6* and *IL8* (encoding IL-6 and IL-8, respectively), also correlated positively in a subcutaneous depot. Increased expression of all these interleukins in adipose tissues of obese individuals has been previously reported [31,32,33]. This finding is not apparent, since in T cells and monocytes, VD was found to down-regulate expression of genes encoding pro-inflammatory cytokines, while upregulating those with anti-inflammatory properties [34]. However, in a culture of mature human adipocytes, an opposite trend was observed that is consistent with our results [7]. Disclosing if the observed correlations between *VDR* and pro-inflammatory interleukins’ mRNA levels in adipose tissues of obese individuals are indeed associated or if they are a coincidence requires further study. 

This is, to our knowledge, the first study trying to assess whether obesity-related changes in vitamin D metabolism affect the severity of local inflammation in adipose tissue. The main limitation of our work is the lack of direct measurements of VD metabolites levels in adipose tissue samples. These data might be crucial to understand the mechanisms responsible for the observed alternations in VD-related genes expression. However, assessment of VD derivates in adipose tissue is challenging from the methodological point of view and the related literature scare [35]. Moreover, since we performed our experiments on adipose tissue homogenates, differences in macrophages infiltration might have influenced the obtained results. However, the observed expression patterns of investigated VD-related genes are similar to those obtained in cultures of primary adipocytes, suggesting that our results are reliable. Another limitation of this study results from the fact that measurements of genes expression were performed on mRNA level. Even though our findings are coherent with the previously published [8,24,25], they might not translate to the protein level. Next, the observed negative correlations between *VDR* mRNA level and expression of the selected miRNAs in obese study participants were limited to the visceral adipose tissue depot, and do not explain obesity-associated changes in *VDR* expression in SAT. Finally, the limited number of the investigated tissues could have also influenced the obtained results. Therefore, further studies performed on the larger numbers of individuals and tissues, followed by in vitro experiments, are necessary to elucidate the role of VD and its receptor in the development of the obesity-associated inflammation in adipose tissue.

In summary, we observed an obesity-associated up-regulation of *VDR* and down-regulation of *CYP27B1* mRNA levels in adipose tissue, which were not associated with VD status of the investigated individuals. On the other hand, we observed a correlation between the mRNA levels of *VDR* in VAT of obese individuals and the expression of the relevant miRNAs. Although our findings were limited to visceral fat depots, based on them, we hypothesize that miRNAs may play a decisive role in the obesity-associated changes in *VDR* expression. Finally, finding that, in obesity, *VDR* mRNA concentrations correlate positively with mRNA levels of pro-inflammatory interleukins may suggest an involvement of local VD status in the development of local inflammation in adipose tissue, however further studies are essential to confirm this observation.

## 4. Materials and Methods

### 4.1. Studied Groups and Tissues

Study participants were recruited from the Department of General and Transplantation Surgery, Medical University of Warsaw, as described previously [19,20,31]. The studied group consisted of 55 patients (48 females and 7 males) diagnosed with 3rd-degree obesity (body mass index, BMI, >40 kg/m^2^), whose basic clinical characteristics, biochemical parameters, and co-morbidities are summarized in Table 1. In this group, patients with pre-diabetes received metformin, while all patients suffering from type 2 diabetes mellitus were treated either with metformin or with a combination of metformin and sulphonylureas. None of the diabetic study participants were using insulin and/or incretin mimetics and/or sodium-glucose transport protein 2 inhibitors. Patients with hypertension received from one to four antihypertensive drugs (angiotensin-converting enzyme (ACE) inhibitor, angiotensin II receptor antagonist, diuretic, calcium channel blocker, or β-blocker). Individuals diagnosed with hyperlipidemia received statins or fibrates.

The control group consisted of 31 normal-weight patients (22 females and 9 males with a BMI of 20–24.9 kg/m^2^) undergoing elective surgical procedures. Apart from cholelithiasis or inguinal hernia, they had no history of any chronic disease, including those associated with obesity and did not receive any pharmacological treatment. Based on their medical history, physical examination and blood tests (Table 1), they were considered to be metabolically healthy. None of the study participants supplemented vitamin D.

Visceral (VAT-O, *n* = 55) and subcutaneous (SAT-O, *n* = 55) adipose tissues were collected from obese patients during bariatric surgery. Control tissues were collected from the patients undergoing elective cholecystectomy (26 samples of VAT-N and 21 samples of abdominal SAT-N) or operated on for inguinal hernia (5 samples of abdominal SAT-N). All samples after collection were immediately frozen at −80 °C and homogenized in liquid nitrogen. 

Before surgery, all participants gave 15 mL of venous blood. Sera obtained from these samples were stored at −80 °C in the dark, in non-permeable tubes, until the measurements were performed.

The study was approved by the Bioethics Committee of the Medical University of Warsaw (decisions no. KB 147/2009—decision date 28 July 2009, KB 91/A/2010—decision date 28 July 2010 and KB 117/A/2011—decision date 26 July 2011) and written informed consent was obtained from all participants.

### 4.2. Measurement of 25(OH)D_3_ and 1,25(OH)_2_D_3_ Concentrations in Sera 

The 25(OH)D_3_ serum concentration was determined using the Elecsys^®^ Vitamin D total II tests (Roche Diagnostics, Rotkreuz, Switzerland) based on high-sensitivity electrochemiluminescence (ECL). According to the manufacturer protocol, the reference values for 25(OH)D_3_ are 27.7–107 nmol/L (11.1–42.9 ng/mL).

The concentration of 1,25(OH)_2_D_3_ was determined according to the AC-62F1 procedure by IDS Immunodiagnostic Systems Ltd. (Frankfurt am Main, Germany), consisting of two stages: initial preparation of serum samples and the proper immunochemical test. In the first stage, serum samples were delipidated with dextran sulphate solution with the addition of magnesium chloride. Then, the serum was incubated with a calcitriol-binding protein inhibitor in columns containing a monoclonal antibody against 1,25(OH)_2_D_3_ bound to the solid phase gel. After the incubation, the columns were rinsed and 1,25(OH)_2_D_3_ was eluted with ethanol. In the second step, an enzyme-linked immunoassay was used to determine the concentration of the purified 1,25(OH)_2_D_3_. The sensitivity range of the method was 6–500 pmol/L (2.52–210 pg/mL).

### 4.3. RNA Isolation, Reverse Transcription and Real-Time PCR

Total RNA was extracted from the homogenized tissues with TRIzol Reagent (Invitrogen, Carlsbad, CA, USA) according to the manufacturer’s protocol. For mRNA analysis, reverse transcription was performed with RevertAid First Strand cDNA Synthesis Kit (Fermentas, Vilnius, Lituania) and the obtained cDNA was used as a template in real-time PCR performed in LightCycler 480 Instrument II (Roche, Mannheim, Germany) with LightCycler 480 Sybr Green I Master Kit (Roche, Mannheim, Germany) and specific primers. The real-time PCR conditions are summarized in Appendix A. The miRNA analysis was performed with a miRCURY™ LNA™ Universal RT microRNA PCR system (Exiqon, Vedbaek, Denmark), as described previously [19,20]. All measurements were performed in triplicate. The results, normalized against the expression of the β-actin gene (*ACTB,* for mRNA) and hsa-miR103a-3p (the recommended control miRNA for the adipose tissue [36]), were presented in arbitrary units (AU) as mean mRNA/miRNA levels.

### 4.4. Statistical Analysis

The normality of distribution and homogeneity of the variance of the studied parameters were checked with the Shapiro–Wilk and Levene’s tests, respectively. The differences in mRNA and miRNA levels were calculated using the Student’s t/Mann–Whitney U test. Correlations between the quantitative values were analysed with the Spearman correlation test. All statistical analyses were performed with the Statistica software package v.10 (StatSoft, Tulsa, OK, USA) and GraphPad Prism software v.7 (GraphPad Software, San Diego, CA, USA).

## Figures and Tables

**Figure 1 ijms-20-05272-f001:**
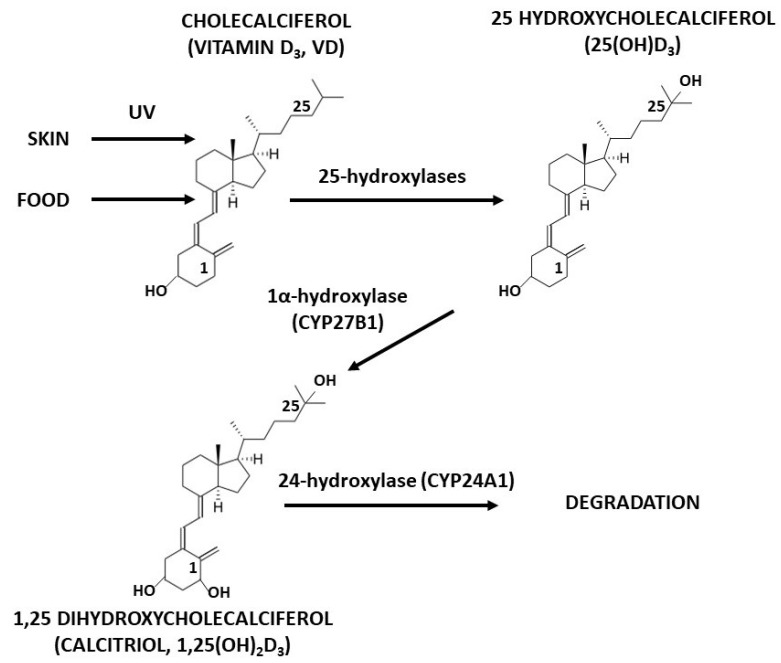
A simplified scheme of vitamin D metabolism. Cholecalciferol (formed under the influence of ultraviolet (UV) rays in the skin or supplied by the diet) undergoes hydroxylation at position 25 in the liver, and then in the reaction catalysed by 1α-hydroxylase (CYP27B1), the active metabolite—1,25-dihydroxycholecalciferol (calcitriol) is formed. Hydroxylation at position 24 (catalysed by vitamin D 24-hydroxylase—CYP24A1) initiates the degradation of vitamin D metabolites.

**Figure 2 ijms-20-05272-f002:**
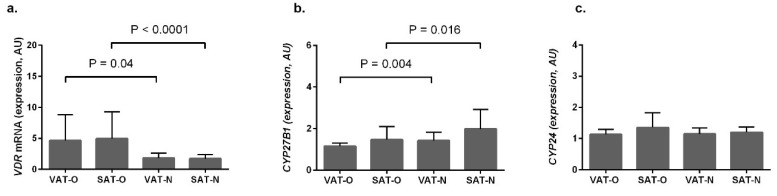
Vitamin D receptor (*VDR*) (**a**) *CYP27B1* (**b**) and *CYP24A1* (**c**) mRNA levels in visceral (VAT) and subcutaneous (SAT) adipose tissue samples from the obese (O) and normal-weight (N) individuals. Results, normalized against the expression of the β-actin gene (ACTB), are presented in arbitrary units (AU) as mean mRNA levels.

**Figure 3 ijms-20-05272-f003:**
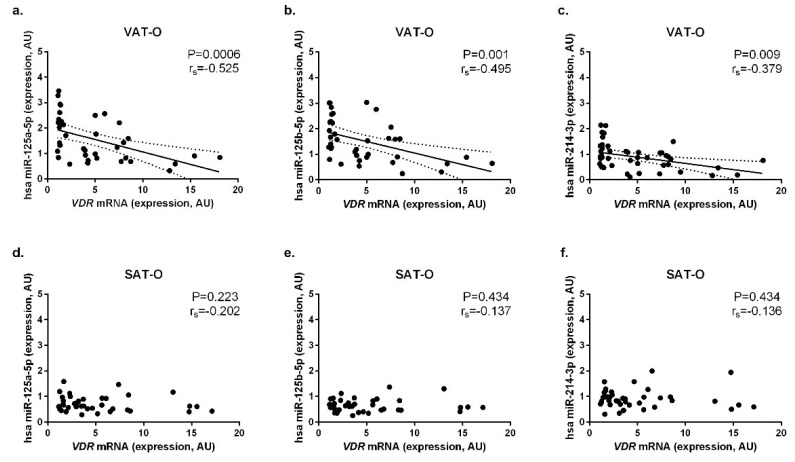
Correlation between mRNA levels of *VDR* and expression of hsa-miR-125a-5p, hsa-miR-125b-5p and hsa-miR-214-3p in visceral (VAT **a**–**c**, respectively) and subcutaneous (SAT **d**–**f**, respectively) adipose tissues of obese (O) individuals.

**Figure 4 ijms-20-05272-f004:**
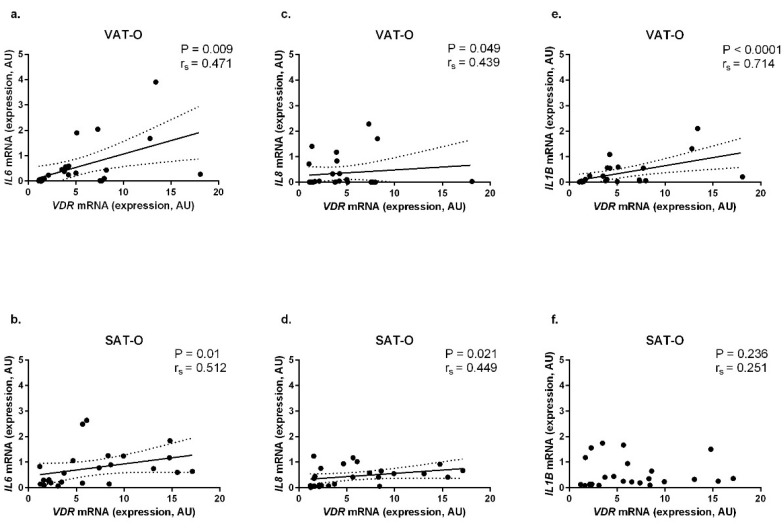
Correlation between mRNA levels of *VDR* and mRNA levels of genes encoding interleukin 6 (*IL6*
**a**,**b**), interleukin 8 (*IL8*
**c**,**d**) and interleukin 1β (*IL1B*
**e**,**f**) in visceral (VAT) and subcutaneous (SAT) adipose tissue of obese (O) individuals.

**Table 1 ijms-20-05272-t001:** Clinical and biochemical characteristics of study participants.

Parameter	Obese Individuals (*n* = 55)		Normal-Weight Controls (*n* = 31)		*p*
	Mean ± SD	Min–Max	Mean ± SD	Min–Max	
Age (years)	41.49 ± 10.40	20–59	45.76 ± 14.81	23–62	0.167
Weight (kg)	131.68 ± 18.09	100–198.6	67.71 ± 11.23	50–90	<0.0001
BMI (kg/m^2^)	46.85 ± 4.74	40.35–59.26	23.42 ± 1.66	20.07–24.95	<0.0001
Adipose tissue (% body mass)	47.96 ± 4.99	32,63–57.23	–	–	
Waist circumference (m)	1.23 ± 0.18	97–167	–	–	
CRP (nmol/L)	96.86 ± 45.14	11.43–184.67	72.0 ± 16.45	1.9–90.11	0.228
Glucose (mmol/L)	5.61 ± 1.46	3.38–10.16	5.21 ± 0.22	4.22–5.44	0.677
Total cholesterol (mmol/L)	5.21 ± 1.07	3.13–7.87	4.85 ± 0.20	3.8–4.92	0.845
LDL (mmol/L)	3.27 ± 1.06	1.25–5.64	2.77 ± 018	2.64–2.90	0.486
HDL (mmol/L)	1.21 ± 0.22	0.78–1.79	1.70 ± 0.64	1.24–2.14	0.179
Triglycerides (mmol/L)	1.65 ± 1.31	1.22–7.62	1.32 ± 0.19	1.08–1.46	0.792
TSH (mIU/L)	1.74 ± 0.86	0.33–3.65	1.22 ± 0.18	1.09–1.35	0.506
25(OH)D_3_ (nmol /L)	39.65 ± 30.02	7.5–174.17	41.48 ± 25.23	9–86.1	0.978
1,25(OH)_2_D_3_ (pmol /L)	130.41 ± 46.30	42.7–260.4	133.99 ± 46.29	59.22–214.71	0.638
Co-morbidities					
Type 2 DM/IGT	26 (47.3%)		none		
Hypertension	32 (58.2%)		none		
Hyperlipidemia	34 (61.8%)		none		
Metabolic syndrome *	33 (60.0%)		none		

N: number of subjects; BMI: body mass index calculated as weight (kg) divided by height squared (m^2^); CRP: C-reactive protein; LDL: low-density lipoproteins; HDL: high-density lipoproteins; TSH: thyroid-stimulating hormone; DM: diabetes mellitus; IGT: impaired glucose tolerance.* The metabolic syndrome was diagnosed based on the International Diabetes Federation criteria for the Europeans available at https://www.idf.org.

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
