# Peer review of "Vitamin D Receptor Gene Expression in Adipose Tissue of Obese Individuals is Regulated by miRNA and Correlates with the Pro-Inflammatory Cytokine Level"

_ijms, 2019, doi:10.3390/ijms20215272_

Round 1

Reviewer 1 Report

Thank you very much for allowing me to review the article "Vitamin D receptor gene expression in adipose tissue of obese individuals is regulated by miRNAs and correlates with pro-inflammatory cytokines level. (IJMS-592907-peer-Review September 2019):

This work is about the vitamin D receptor (VDR) and its identification in a wide panel of tissues that improves the knowledge of the role of vitamin D3 (cholecalciferol, hereinafter referred to as vitamin D, VD) in the maintenance of homeostasis bodily. In this context of adipose tissue, active forms of RV participate in the regulation of lipogenesis, adipogenesis, as well as the production and secretion of adipokines. The working hypothesis is that the expressive activity of the CYP27B1, CYP24A1 and VDR genes, all of them with VDRE in their regulatory regions, can be modulated by the RV itself, but also by other factors of which they perform important epigenetic modifications such as microRNA interference (miRNA). For all the above, this study aims to assess whether obesity changes the expression of genes crucial for the activation of RV (CYP27B1), inactivation (CYP24A1) and action (VDR) in adipose tissues and, if so, Observe whether the phenomenon has any influence on the local inflammatory environment. In addition, they assessed whether microRNA interference could play a role in regulating the expression of RV-related genes in adipose tissue.

Review Comments: I believe that the authors must complete the material and methods section that allows identifying the participating individuals, the approval of the Ethics Committee and explaining how they have been recruited for the study.
The other sections I think are clearly presented.
The results are very interesting: they identify that excessive adiposity in humans is associated with changes in the expression of VD-related genes in adipose tissue and seek for the possible mechanisms responsible for this phenomenon my recommendation that this work be completed with the material and methods section so that it can be considered for acceptance.

Author Response

Reviewer #1

“I believe that the authors must complete the material and methods section that allows identifying the participating individuals, the approval of the Ethics Committee and explaining how they have been recruited for the study. The other sections I think are clearly presented.”

 We thank the Reviewer for this comment. In the revised version of the manuscript, we have modified the description of the studied groups following the Reviewer’s suggestion.

“Study participants were recruited from the Department of General and Transplantation Surgery, Medical University of Warsaw, as described previously [19,20,31]. The studied group consisted of 55 patients (48 females and 7 males) diagnosed with 3rd-degree obesity (body mass index, BMI, > 40 kg/m2), whose basic clinical characteristics, biochemical parameters, and co-morbidities are summarized in Table 1. In this group, patients with pre-diabetes received metformin, while all patients suffering from type 2 diabetes mellitus were treated either with metformin or with a combination of metformin and sulphonylureas. None of the diabetic study participants were using insulin and/or incretin mimetics and/or sodium-glucose transport protein 2 inhibitors. Patients with hypertension received from one to four antihypertensive drugs (angiotensin-converting enzyme (ACE) inhibitor, angiotensin II receptor antagonist, diuretic, calcium channel blocker, or β-blocker). Individuals diagnosed with hyperlipidemia received statins or fibrates.

The control group consisted of 31 normal-weight patients (22 females and 9 males with a BMI of 20–24.9 kg/m2) undergoing elective surgical procedures. Apart from cholelithiasis or inguinal hernia, they had no history of any chronic disease, including those associated with obesity and did not receive any pharmacological treatment. Based on their medical history, physical examination and blood tests (Table 1), they were considered to be metabolically healthy. None of the study participants supplemented vitamin D.” (Page 7, lines 262-278)

“The study was approved by the Bioethics Committee of the Medical University of Warsaw (decisions no. KB 147/2009, KB 91/A/2010 and KB 117/A/2011) and written informed consent was obtained from all participants” (Page 8, lines 286-288)

Reviewer 2 Report

The authors conducted clinical study how VD related genes and subsequent inflammatory markers correlate with adipose tissues, subcutaneous and visceral.

The epidemiological and basic research revealed that visceral adipose tissue plays more pivotal role in metabolic syndrome and inflammation than subcutaneous tissue.  However, in the current study, no apparent difference except for miRNA effect on VDR were found between two tissues.  Authors should discuss the clinical implication of the data.

In addition, what is the mechanisms underlie in SAT to regulate VD-related genes besides miRNA?

Figure 3, I could no find figure 3d and data from SAT.

Figure 4, I could no find data on IL1beta from SAT.

Author Response

Reviewer #2

The epidemiological and basic research revealed that visceral adipose tissue plays more pivotal role in metabolic syndrome and inflammation than subcutaneous tissue. However, in the current study, no apparent difference except for miRNA effect on VDR were found between two tissues. Authors should discuss the clinical implication of the data.

              We thank the Reviewer for this valuable suggestion. Indeed, numerous studies (including these from our research group: Jonas et al., 2015; Kurylowicz et al., 2017) prove that adipose tissue depots differ in their metabolic and secretional activity. Subsequently, excess adiposity of different location is associated with different risk of obesity-related complications. Most studies point to the visceral adipose tissue (VAT) as the primary source of unfavourable mediators that impact negatively whole-body function (e.g. Heilbronn et al., 2008; Pou et al., 2007; Alvehus et a., 2010). There are reports, however, that question the dominant role of VAT in the development of obesity-associated inflammation (e.g. Spoto et al., 2015; Jonas et al., 2015). Given the fact that vitamin D (VD) is implicated in the local regulation of the inflammatory process in adipose tissue, finding that in obesity expression of VD related genes, investigated in our study, is similarly up-regulated (in case of VDR) or down-regulated (in case of CYP27B1) in both adipose tissue depots might be somehow surprising. However, our measurements were performed on mRNA level, and we cannot predict how they may translate to the protein level (that was enlisted as one of the limitations of the study). We discussed this fact in the corrected version of the manuscript.

“Interestingly, in obese individuals, we did not observe significant differences in VDR and CYP27B1 expression between the different adipose tissue depots. This finding may reflect the true phenomenon or result from the fact that our measurements were performed on the mRNA level and might not translate to the protein level.” (Page 6 lines 193-197)

“Another limitation of this study results from the fact that measurements of genes expression were performed on mRNA level. Even though our findings are coherent with the previously published [8,23,24], they might not translate to the protein level.”(Page 7 lines 245-248)

In addition, what is the mechanisms underlie in SAT to regulate VD-related genes besides miRNA?

It is a valuable suggestion, and the paragraph discussing the potential mechanisms responsible for the regulation of VD-related genes was added to the manuscript with the proper reference:

“We found significant, negative correlations between VDR mRNA level and levels of hsa-miR-125a-5p, hsa-miR-125b-5p, and hsa-miR-214-3p in VAT of obese subjects. All these miRNAs were found to participate in the regulation of VDR expression in vitro [15,21,22], and their expression is decreased in visceral adipose tissue in obesity [19,20]. We do not observe similar correlations in SAT of obese individuals. Given the numerous differences in metabolism and secretional activity between VAT and SAT, the mechanisms controlling genes expression in these two adipose tissue depots might also be distinct. In the case of VDR, its tissue-specific expression may result, e.g., from the regulation by different enhancers present upstream of the promoter [29] or different methylation levels in the regulatory regions [26].” (Page 6, lines 212-220)

Ref. 29.  Lee, S.M.; Meyer, M.B.; Benkusky, N.A.; O'Brien, C.A.; Pike, J.W. Mechanisms of Enhancer-mediated     Hormonal Control of Vitamin D Receptor Gene Expression in Target Cells. J Biol Chem 2015, 290, 30573-      30586.

“Figure 3, I could no find figure 3d and data from SAT."

We thank the Reviewer for this remark and apologize for the oversight. In the revised version of the manuscript, we have added the data from SAT-O to Figure 3 and changed its legend. We have also prepared a Supplementary Figure 2 presenting the results of correlations between the mRNA levels of VDR and expression of hsa-miR-125a-5p, hsa-miR-125b-5p and hsa-miR-214-3p in visceral and subcutaneous tissues of normal-weight study participants. Additionally, we have prepared a Supplementary Figure 3 presenting the results of correlations between the mRNA levels of CYP27B1 and expression of hsa-miR-22-3p and hsa-miR-450b-5p in visceral and subcutaneous tissues of obese and normal-weight individuals (please see the attachment)

“In obese individuals, we observed significant negative correlations between VDR mRNA level and hsa-miR-125a-5p (P = 0.0006, r­s = −0.525, Figure 3a), hsa-miR-125b-5p (P = 0.001, r­s = −0.495, Figure 3b) and hsa-miR-214-3p (P = 0.009, r­s = −0.379, Figure 3c) in VAT. However, we did not observe similar correlations in SAT of obese individuals or in tissues obtained from the normal-weight subjects (Supplementary Figure 2).” (Page 4, lines 133-137) 

“Among miRNAs targeting CYP27B1, expression of hsa-miR-22-3p and hsa-miR-450b-5p differed significantly between the investigated tissues (Supplementary Figure 1, Kurylowicz A, 2016 and 2017 [19,20]. However, we observed no significant correlations between the expression of these miRNA and CYP27B1 mRNA levels (Supplementary Figure 3).” (Page 4, lines 143-146)

Figure legends:

Figure 3. Correlation between mRNA levels of VDR and expression of hsa-miR-125a-5p, hsa-miR-125b-5p and hsa-miR-214-3p in visceral (VAT a, b, c, respectively) and subcutaneous (SAT d, e, f, respectively) adipose tissues of obese (O) individuals. (Page 4, lines 140-146)

Supplementary Figure 2. Correlation between mRNA levels of VDR and of hsa-miR-125a-5p, hsa-miR-125b-5p and hsa-miR-214-3p in visceral (VAT a, b, c, respectively) and subcutaneous (SAT d, e, f, respectively) adipose tissues of normal-weight (N) individuals.

Supplementary Figure 3. Correlation between mRNA levels of CYP27B1 and expression of hsa-miR-22-3p (a, b, c, d) and hsa-miR-450b-5p (e, f, g, h) in in visceral (VAT) and subcutaneous (SAT) adipose tissues of obese (O) and normal-weight (N) individuals.

“Figure 4, I could no find data on IL1beta from SAT.”

We thank the Reviewer for this remark and apologize for the oversight. In the revised version of the manuscript, we have added the data regarding the correlation between mRNA levels of VDR and mRNA levels of IL1B in subcutaneous adipose tissue of obese study participants (Figure 4f). Subsequently, Figure 4 legend has been changed:

Figure 4. Correlation between mRNA levels of VDR and mRNA levels of genes encoding interleukin 6 (IL6 a, b), interleukin 8 (IL8 c, d) and interleukin 1β (IL1B e, f) in visceral (VAT) and subcutaneous (SAT) adipose tissue of obese (O) individuals. (Page 5, lines 156-158)

Reviewer 3 Report

Although this reviewer warmly welcomes this manuscript, some questions should be addressed:

A major drawback of this work is the lack of a clear guideline and aim that the Reader can follow.

The functional mechanisms linking fat, endothelial function, and metabolism (J Clin Hypertens. 2019;21(2):239-242) should be better addressed.

The key role of inflammation in cardiovascular risk (Gambardella J et al. Atherosclerosis. 2016 Oct;253:258-261) should be extensively discussed.

Improve the resolution of Figure 1.

The strengths and limitations of the study should be deeply addressed, taking into account sources of potential bias or imprecision: Discuss both direction and magnitude of any potential bias.

English language (syntax, grammar, correct choice of words, correct use of adjectives and adverbs) should be substantially improved throughout the text.

Author Response

Reviewer #3

A major drawback of this work is the lack of a clear guideline and aim that the Reader can follow.

We thank the Reviewer for this valuable suggestion. Indeed, the part of the Introduction defining the goals of the study was not clear. With this in mind, we have modified this section, hoping that now it will be more transparent for the potential reader.

“In this study, we aimed to evaluate if obesity influences the expression of genes crucial for VD activation (CYP27B1), inactivation (CYP24A1) and action (VDR) in different adipose tissues depots. Next, we investigated if VD status and miRNA interference could play a role in this phenomenon. Finally, by correlating the mRNA levels of these genes with mRNA levels of the pro-inflammatory cytokines, we aimed to assess how obesity-related changes in VD-related genes expression may impact the local inflammatory milieu in adipose tissue.” (Page 3, lines 84-90)

We have also modified the Results section so that subsequent sections correspond to the specific objectives of the study:

Results

2.1. VDR and CYP27B1 have opposite expression profiles in adipose tissues

2.2. VDR and CYP27B1 expression profiles do not depend on vitamin D metabolites’ serum levels

2.3. Expression of the selected miRNAs correlates negatively with VDR mRNA levels in adipose tissues

2.4. VDR mRNA levels correlate positively with the expression of genes encoding pro-inflammatory cytokines in adipose tissues of obese individuals

The functional mechanisms linking fat, endothelial function, and metabolism (J Clin Hypertens. 2019;21(2):239-242) should be better addressed. The key role of inflammation in cardiovascular risk (Gambardella J et al. Atherosclerosis. 2016 Oct;253:258-261) should be extensively discussed.

Following the Reviewer’s suggestion, we added a paragraph into the Introduction discussing the impact of inflammatory mediators secreted by adipose tissue on the development of obesity-related cardiovascular complications with proper references.

“In the course of obesity, the accumulation of lipids leads to the increased expression of genes encoding cytokines, chemokines and adhesion molecules in adipocytes, leading to the infiltration of immune cells that produce further pro-inflammatory mediators [11]. This shift in the secretory profile of adipose tissue contributes to the development of obesity-related metabolic and cardiovascular complications [12]. Increased levels of pro-inflammatory adipokines circulating in the blood impair endothelial function and lead to the thickening of the vascular smooth muscle layer. These subsequently result in increased oxidative stress, stiffening of the vascular wall and impaired vessel reactivity [13]. The unfavourable effect of adipokines on the cardiovascular system can also be exerted indirectly, via other organs and systems, e.g., obesity-associated high leptin serum levels increase blood pressure by the stimulation of certain pathways in the central nervous system and by enhancing the sympathetic nervous system output [12].” (Page 2, lines 54-64)

Ref. 12.  Shu, J.; Matarese, A,; Santulli, G. Diabetes, body fat, skeletal muscle, and hypertension: The ominous chiasmus? J Clin Hypertens (Greenwich) 2019, 21, 239-242.

Ref. 13. Gambardella, J.; Santulli, G. Integrating diet and inflammation to calculate cardiovascular risk. Atherosclerosis 2016, 253, 258-261.

Improve the resolution of Figure 1.

Following the Reviewer's suggestion, we improved the resolution of Figure 1.

The strengths and limitations of the study should be deeply addressed, taking into account sources of potential bias or imprecision: Discuss both direction and magnitude of any potential bias.

It is a valuable suggestion, and the paragraph discussing the strengths and limitations of the study was added to the discussion section.

“This is, to our knowledge, the first study, performed on a representative number of tissues, trying to assess whether obesity-related changes in vitamin D metabolism affect the severity of local inflammation in adipose tissue. The main limitation of our work is the lack of direct measurements of VD metabolites levels in adipose tissue samples. These data might be crucial to understand the mechanisms responsible for the observed alternations in VD-related genes expression. However, assessment of VD derivates in adipose tissue is challenging from the methodological point of view and the related literature scare [35]. Moreover, since we performed our experiments on adipose tissue homogenates, differences in macrophages infiltration might have influenced the obtained results. However, the observed expression patterns of investigated VD-related genes are similar to those obtained in cultures of primary adipocytes, suggesting that our results are reliable. Another limitation of this study results from the fact that measurements of genes expression were performed on mRNA level. Even though our findings are coherent with the previously published [8,24,25], they might not translate to the protein level. Finally, the observed negative correlations between VDR mRNA level and expression of the selected miRNAs in obese study participants were limited to the visceral adipose tissue depot, and do not explain obesity-associated changes in VDR expression in SAT.” (Pages 7, lines 236-251).

“English language (syntax, grammar, correct choice of words, correct use of adjectives and adverbs) should be substantially improved throughout the text.”

                  Following the Reviewer suggestion, we took the advantage from the English editing service provided by MDPI (see the attached certificate).

Round 2

Reviewer 1 Report

Thank you very much for allowing me to review, on a second occasion, the article "Vitamin D receptor gene expression in adipose tissue of obese individuals is regulated by miRNAs and correlates with pro-inflammatory cytokines level. (IJMS-592907-Review September 2019):

Review Comments: After reviewing the article and the annexes, I have verified that the authors have improved the introduction, have also readjusted the objectives.
The authors have completed the material and methods.

However, in the discussion I believe that the weakness of a study with such a small sample size on VDR should be included, since it is related to and interacting with multiple pathologies and medications, so that larger studies are necessary and if possible Follow-up studies to confirm the results obtained.

Reviewer 2 Report

The authors partly revised and no data are missing in the current version.  Hoever, I found serious issues in data and I doubt the integrity of data collection and analysis.

Judging from the method, authors used same data sets for Figures 2, 3, and f for VDR mRNA, but the distribution of VDR mRNA apparently different between Figure 3 and 4.  See SAT-O, in figure 3 VDR mRNA distribute from 0 to 10 AU, but in figure 4, it distribute 0 to 20.  In Figure 2 mean VDR mRNA of VAT-O and SAT-O are almost equal and mean value is around 4 AU.  Neither Figure 3 nor 4 shows the mean values of VDR mRNA from VAT-O or SAT-O are around 4 AU.

The same kind of concern is raised for hsa-mir 125a,b and 214 in figures 3, supple figure 1.  Mean hsa-miR-125a from VAT-O is around 1 in supple figure 1 but it is apparently larger than 1 in Figure 3a.  So as for 125b.

Again for CYP27B1, supple Figure 3 it of VAT-O varies from 1.0 to 1.6 AU but the mean of CYP27B1 in Figure 2 is around 1 AU.

Above those concerns are serious and I can not accept the article.

Reviewer 3 Report

-

Author Response

We thank the Reviewer for accepting the revisions and the time spent on reviewing our manuscript.

Round 3

Reviewer 2 Report

The authors followed my suggestion and data are clearly shown in this current version, however, I have concern p value of statistical analysis.

Authors should double check if p value are correct.  Judging from data provided, I calculated p value and some are widely differ from the presented p value.